# Recent Advances in Our Molecular and Mechanistic Understanding of Misfolded Cellular Proteins in Alzheimer’s Disease (AD) and Prion Disease (PrD)

**DOI:** 10.3390/biom12020166

**Published:** 2022-01-20

**Authors:** Walter J. Lukiw

**Affiliations:** Departments of Neurology, Neuroscience and Ophthalmology, LSU Neuroscience Center, Louisiana State University Health Sciences Center, New Orleans, LA 70112, USA; WLukiw@lsuhsc.edu; Tel.: +504-599-0842

**Keywords:** Alzheimer’s disease (AD), atypical protein folding, frontotemporal degeneration (FTD), prion disease (PrD), proteostasis, tauopathies

## Abstract

Naturally occurring neuron-abundant proteins including amyloid Aβ42 peptide and the microtubule-associated protein tau (MAPT) can, over time and under pathological situations, assume atypical conformations, altering their normal biological structure and function, and causing them to aggregate into insoluble and neurotoxic intracellular inclusions. These misfolded proteins ultimately contribute to the pathogenesis of several progressive, age-related and ultimately lethal human neurodegenerative disorders. The molecular mechanism of this pathological phenomenon of neuronal protein misfolding lends support to the ‘*prion hypothesis*’, which predicts that the aberrant folding of endogenous natural protein structures into unusual pathogenic isoforms can induce the atypical folding of other similar brain-abundant proteins, underscoring the age-related, progressive nature and potential transmissible and spreading capabilities of the aberrant protein isoforms that drive these invariably fatal neurological syndromes. The abnormal folding and aggregation of host proteins is a consistent feature of both amyloidopathies and tauopathies that encompass a continuous spectrum of brain diseases that include Alzheimer’s disease (AD), prion disorders (PrD) such as scrapie in sheep and goats (*Bovidae*), experimental prion infection of rodents (*Muridae*), Creutzfeldt–Jakob disease (CJD) and Gerstmann–Sträussler–Scheinker syndrome (GSS) in humans (*Hominidae)*, and other fatal prion-driven neurological disorders. Because AD patients accumulate both misfolded tau and Aβ peptides, AD may be somewhat unique as the first example of a ‘*double prion disorder*’. This commentary will examine current research trends in this fascinating research area, with a special emphasis on AD and PrD, and the novel pathological misfolded protein processes common to both intractable neurological disorders.

## 1. Introduction

Translation of the genetic information encoded in messenger RNA (mRNA) by ribosomes is an extremely rapid biological process currently estimated to be about ~16.8 nucleotides (nt) being read per second, with an incorporation and elongation rate of the growing polypeptide chain of about ~5.6 amino acids per second [1,2,3]. Polyribosome-based reading of single mRNAs involving both circular and linear mRNA–ribosome configurations can considerably amplify the rate of this process [4,5]. The rapid primary, secondary, tertiary and other three-dimensional folding of amino acids into nascent polypeptide chains occurs simultaneously with amino acid polymerization within the ribosomal cleft immediately after their biosynthesis [2,3,5,6]. The overall process of rapid translation involving the formation of biologically active protein structures has emerged and evolved to maintain cellular protein homeostasis, sometimes referred to as proteostasis, by diminishing the possibility of both polypeptide misfolding and the self-aggregation of newly formed polypeptides, thus ensuring the maintenance and functionality of a healthy and optimally performing cellular proteome [5,6,7,8,9].

Unusual and atypical protein folding and the self-association of polypeptides into insoluble aggregates is highly problematic for homeostatic protein structure and function and cellular proteostasis, with ensuing deleterious biological consequences [9,10]. There is increasing evidence that multiple progressive and age-related human neuropsychiatric and neurodegenerative disorders, including Alzheimer’s disease (AD), frontotemporal dementia (FTD), Parkinson’s disease (PD), progressive supranuclear palsy (PSP), prion disease (PrD) and other invariably lethal neurological disorders, are the result of: (**i**) an altered folding of host-encoded cellular proteins that is closely associated with their self-aggregation into disease-linked pathological lesions within the brain and central nervous system (CNS), and (**ii**) a defective autophagy that normally serves as a quality control system by which dysfunctional cytoplasmic components, protein aggregates and/or damaged organelles are degraded by lysosomal surveillance systems [10,11]. Highly relevant to these observations by many laboratories is the ‘*prion hypothesis*’, which predicts that the abnormal folding of endogenous natural host protein structures into unusual pathogenic isoforms can, over time, induce the atypical folding and aggregation of other similar brain-abundant proteins, and underscores both the age-related and progressive nature of these lethal human neurological disorders [9,10,12]. 

## 2. Amyloid Beta (Aβ) Peptides 

The two major histopathological hallmarks of AD are the formation of amyloid-beta (Aβ) peptides containing amyloid plaques and neurofibrillary tangles (NFTs) composed of altered tau proteins; these are accompanied by a progression in neuronal atrophy and the loss of neurons and synapses, the latter of which correlates best with cognitive impairment in the AD patient [13,14,15,16,17]. The 695–751 amino acid beta-amyloid precursor protein (βAPP; encoded at human chromosome 21q21.3; https://www.genecards.org/cgi-bin/carddisp.pl?gene=APP; last accessed 18 January 2022), a Type 1 integral trans-membrane protein, is endoproteolytically processed to generate either the neurotrophic secreted ectodomain constituting APP alpha (sAPPα) or a series of ragged, neurotoxic, amyoidogenic amyloid beta (Aβ) peptides [18,19]. The 40–42 amino acid amyloid-beta (Aβ40, Aβ42) peptides and other oligomeric Aβ species are derived from the sequential tandem endoproteolytic cleavage of βAPP by the amyloidogenic beta- and gamma-secretases [18,19,20,21]. While the neurobiology and neurophysiology of βAPP holoprotein is incompletely understood, βAPP’s most recent functional assignments include essential roles in cellular adhesion, neural growth and repair, neurogenesis, synaptic plasticity, neurite outgrowth and/or neuroprotection, and many of these neurotrophic functions are moderated by the α-secretase-mediated βAPP ectodomain cleavage into sAPPα [17,18,20,21]. 

Similarly, recent evidence has suggested that different isoforms of Aβ peptides may have a normal function as innate-immune system-activated immune-peptides possessing a neuromodulatory, neuroprotective and/or antimicrobial role [22,23,24]. On the other hand Aβ40 and Aβ42 peptides and related monomeric or oligomeric Aβ species that differ in molecular weight, conformation and morphology are fundamental in promoting and sustaining the formation of polymorphic, neurotoxic and pro-inflammatory Aβ-peptide containing globular and fibrillar structures. These structures represent the pathological basis for driving amyloid deposition and the Aβ aggregation pathway, pro-inflammatory signaling, and the maintenance and propagation of AD progression over time [22,25]. The reaffirmed capabilities for Aβ peptides to promote tau seeding and aggregation, and the cross-talk between the two basic lesions that define AD suggest a specific and plausible mechanism by which extracellular Aβ peptides initiate the neuro-destructive pathological cascade that is unique to AD and its unique classification as a ‘*double prion disorder*’ [12,26,27,28,29].

## 3. Microtubule-Associated Protein Tau (MAPT) Proteins

Neuron-abundant ~352 to ~441 amino acid microtubule-associated protein tau proteins (MAPT; chr 17q21.31; https://www.genecards.org/cgi-bin/carddisp.pl?gene=MAPT; last accessed 18 January 2022) are normally involved in the organization and stabilization of the internal microtubule system of cells of the human CNS. Recently described as ‘*intrinsically disordered*’, MAPT proteins are centrally involved in homeostasis, the maintenance of neuronal cytoarchitecture, three-dimensional neuronal shape and overall synaptic organization [30,31,32]. Interestingly MAPT gene transcripts in the human brain and CNS undergo alternative splicing from a single gene to yield six different tau protein isoforms that are expressed at different ratios in neurodegeneration, which result in tau pathology of paired-helical filaments, neurofibrillary tangles, and tau fibrillar aggregates with detrimental effects such as microtubule destabilization [16,33].

Similar to the pathological self-association of Aβ40 and Aβ42 peptides, MAPT proteins also self-aggregate into complex, intracellular inclusions. and contribute to aggregated proteomes, pro-inflammatory responses and neurotoxicity in a number of different progressive, age-related and ultimately fatal neurodegenerative diseases termed ‘tauopathies’ [16,27,33,34]. The tauopathies consists of at least two dozen dissimilar, invariably lethal and age-related neurological disorders that include, prominently, Alzheimer’s disease (AD), argyrophilic grain disease (AGD), chronic traumatic encephalopathy (CTE), corticobasal degeneration (CBD), Parkinson’s disease (PD), Pick’s disease (PiD) and progressive supranuclear palsy (PSP), as well as several other more rare tauopathies exhibiting clinical symptoms that include frontotemporal dementia, Richardson syndrome, FTLD-tau (frontotemporal dementia with Parkinsonism caused by MAPT mutations), pure akinesia with gait freezing and motor neuron symptoms and/or cerebellar ataxia, familial British dementia, familial Danish dementia, primary age-related tauopathy, and several other more rare tau-mediated disorders, as well as a growing number of prion or prion-like disorders [12,32]. 

## 4. Elucidation of the Organization of Aβ and MAPT Proteins in Neurological Health and Disease

Very recently, significant scientific progress has been made in our conceptualization of the pathological mechanism by which unusually folded and self-associating tau proteins drive the neuropathology of several common types of human neurodegenerative disease. Each of the tauopathies are pathologically defined by the progressive accumulation in the brain of MAPT proteins as complex fibrillar aggregates, and their incidence and prevalence are now known to correlate strongly with the degree of dementia [33,35]. Interestingly, dominant tau mutations cause both increased tau aggregation, neuro-inflammation and neurodegeneration, and the pathogenesis of different tauopathies appears to involve pathological tau conformations that serve as templates that recruit native tau proteins to form additional abnormally folded tau proteins that support the further generation of self-aggregating assemblies. These molecular mechanisms are extraordinarily similar to those involving the generation, propagation, diffusion and spreading of prions, and have implications for the potential ‘seeding’ and horizontal transmission of the entire spectrum of human misfolded tau- and α-synuclein-linked diseases, which include AD and Parkinson’s disease (PD) [12,34,36].

Cryo-electron microscopy (cryoEM) is an advanced electron microscopic technique that typically involves freezing a biological sample in an ethane slush that produces a vitreous or ‘non-crystalline’ ice; the frozen sample grid is then maintained at −198 °C (liquid nitrogen temperature) enabling the viewing and digitized imaging with the least possible distortion and the fewest possible artifacts normally generated by ‘crystalline’ forms of ice [37]. While data acquisition, image processing and averaging of multiple images provide high-resolution information of considerably less than ~1 nm (atomic resolution), another major advantage of cryoEM technology over traditional EM includes the preservation of the biological sample in a near-native hydrated state without the distortions induced by stains or fixatives needed for traditional EM [37] (https://www.nature.com/articles/d41586-020-02924-y; last accessed 18 January 2022).

Utilizing data derived from cryo-EM technologies, the Scheres laboratory at the MRC Laboratory of Molecular Biology, Cambridge, UK, have recently described a basic hierarchical classification of tauopathies that can be made on the basis and nature of the unusually folded structuring of tau filaments in each disease type [35]. This in itself is a major advancement in our understanding of the organization and classification of molecular variations associated with human tau-linked neurological disease [12,35,37].

Interestingly, of the six tau isoforms naturally expressed in the human brain, three of these isoforms have three microtubule-binding repeat (MTR) domains, and three isoforms have four MTR domains [16,35,38]. In general, the tauopathies can currently be classified into several basic groups based on the isoforms that constitute the abnormal tau filaments; for example, AD, CTE, PiD and CBD are each characterized by different tau 3MTR- or 4MTR-type molecular folds [12,33,38]. Because of the unique and characteristic topology and topography of the 3MTR- and/or 4MTR-folded structures, it should be possible to create a series of highly specific antibodies that discriminate between these various kinds of molecular folds, and hence the different types of tauopathies, which, to date, have been exceedingly difficult to analyze, evaluate and diagnose in the neurology clinic. This important classification system should advance both diagnostic and prognostic methods for the early detection of neurodegeneration that will facilitate preclinical trials for experimental pharmaceuticals, but should also aid in the clinical management of these progressive prion-like diseases, which currently have neither any effective treatment nor cure.

Taken together, this recent, novel, technology-driven classification of tau-associated neurological disorders: (**i**) has significantly expanded our understanding and appreciation of the diversity and variety in the onset and clinical presentation of the continuum of these age-related neurological diseases; (**ii**) has illustrated the immense potential of the directed application of cryo-EM to the elucidation of lethal neurological diseases involving misfolded proteins and abnormal protein conformations; and (**iii**) has underscored the close molecular and structural interrelationships amongst the many different types of human tauopathies. Major medical, scientific and technological challenges are to conceive and implement standardized analytical approaches, techniques and protocols to accurately categorize neurological disorders based on the pathological molecular structure of the underlying abnormal tau protein assemblies. This will achieve an increased accuracy in the diagnosis, prognosis and effective therapies for the clinical management of the wide spectrum of neurological diseases involving misfolded MAPT, Aβ peptides and the related neuronal-enriched fibrillar proteins, and their closely related potential for self-aggregation, as well as the seeding, progression and propagation of neuropathological events [12,35,36,39]. 

## 5. Summary

Our molecular and mechanistic understanding of the contribution of specifically misfolded cellular amyloid and MAPT proteins to the neuropathology of AD and/or PrD continues to evolve. The identification and appreciation of these complex molecular processes remains both enigmatic and fascinating for a number of reasons. These include: (**i**) the implication that both of these progressive age-related disorders are, in a large part, based on the abnormal folding of endogenous natural protein structures into unusual pathogenic isoforms that, over time, can stimulate the atypical folding of other natural brain-enriched proteins; (**ii**) that this induction of misfolded pathogenic proteins from normal proteins may be partly based on their underlying primary amino acid sequence, thus defining a set of biological and biochemical parameters which favor such a pathological conformational transition in certain types of brain proteins; (**iii**) that these novel and unique disease processes, notably without the direct involvement of nucleic acids, also provide a mechanistic foundation for ‘seeding’ or disease spreading that may partly occur via leakage of these peptides and/or proteins through normally protective biological barriers or perhaps via intra- or inter-cellular signaling via exosomes and/or through extracellular micro-vesicular transport [39,40]; (**iv**) the implication that the rate of abnormal protein folding may be augmented and/or amplified by other neurotoxic pathological factors, environmental toxins and/or processes that alter the native polypeptide conformation to ultimately promote protein misfolding; (**v**) the general inference that these pathological processes are extraordinarily similar to those involving the generation, propagation, diffusion and spreading of prions in multiple mammalian neurological disorders [28,29]; and (**vi**) the intriguing possibility that specific abnormally folded brain proteins have pivotal roles in the initiation, propagation, and perhaps the ‘seeding’ and/or ‘horizontal transmission’ within the entire spectrum of misfolded MAPT-, α-synuclein- and/or Aβ peptide-linked neurodegenerative diseases observed to occur in multiple susceptible mammalian species [12,34,36,39].

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
