# Peer review of "Recent Advances in Our Molecular and Mechanistic Understanding of Misfolded Cellular Proteins in Alzheimer’s Disease (AD) and Prion Disease (PrD)"

_biomolecules, 2022, doi:10.3390/biom12020166_

Round 1

Reviewer 1 Report

This commentary from Professor Lukiw compiles data from Alzheimer’s disease as a prion-like disease due to protein misfolding.  It is well written and relevant in the field. Only two minor concerns to be addressed: 

Line 23: "scrapie in rodents" is not correct, while scrapie has been experimentally studied in rodents it is a prion disease from sheep and goats.

Line 119: "prion disorders" are directly related to the misfolding of the prion protein. Other disorders sharing the accumulation of misfolded pathological proteins -like those mentioned in this paragraph" are known as "prion-like disorders".  

Author Response

RESPONSES TO REVIEWER #1:

First of all we would like to thank Reviewer 1 for their valuable time and expertise in the review of our work. Each of their comments below are followed by our ‘RESPONSE’. We feel that these changes, additions and clarifications to our ‘Commentary’ have resulted in a strengthened contribution to this special Issue of Biomolecules.

COMMENT: Line 23: "scrapie in rodents" is not correct, while scrapie has been experimentally studied in rodents it is a prion disease from sheep and goats.

RESPONSE: Thank you for this important point. This has been corrected in the revised manuscript text. The changed text now reads:

‘The abnormal folding-and-aggregation of host proteins is a consistent feature of both amyloidopathies and tauopathies that encompass a continuous spectrum of brain disorders that include Alzheimer’s disease (AD), prion disorders (PrD) such as scrapie in sheep and goats (Bovidae), experimental prion infection of rodents (Muridae) and Creutzfeldt–Jakob disease (CJD) and Gerstmann–Sträussler–Scheinker syndrome (GSS) in humans (Hominidae), and other fatal prion-driven neurological disorders’.

COMMENT: Line 119: "prion disorders" are directly related to the misfolding of the prion protein. Other disorders sharing the accumulation of misfolded pathological proteins -like those mentioned in this paragraph" are known as "prion-like disorders".

RESPONSE: Thank you again for pointing this out this important point. This has been corrected in the revised manuscript text.

Reviewer 2 Report

This is an interesting and timely report.  There are just a few minor areas for improvement that should be considered.

  1. “…have recently reported the basic formulation for a  hierarchical classification of tauopathies that can be made on the basis and nature of the unusual folded structuring of tau filaments in each disease type”.  The meaning of this is obscure is should be elaborated on and its importance described, if it is to be included.
  2. A little more discussion of why irresolvable and inert proteinaceous inclusions in the cell seem to be so harmful would be useful.
  3. A few statements consist largely of ‘filler” without actually moving things forward. For example “Our molecular and mechanistic understanding of the contribution of specifically mis-folded cellular amyloid and MAPT proteins to AD and/or PrD neuropathology continues to evolve…”  Cutting down on these should be considered.

Author Response

RESPONSES TO REVIEWER #2:

First of all we would like to thank Reviewer 2 for their valuable time and expertise in the review of our Commentary. Each of the comments below are followed by our ‘RESPONSE’. We feel that these changes, additions and clarifications to our ‘Commentary’ article have resulted in a strengthened contribution to this special Issue of Biomolecules.

COMMENT: “…have recently reported the basic formulation for a  hierarchical classification of tauopathies that can be made on the basis and nature of the unusual folded structuring of tau filaments in each disease type”.  The meaning of this is obscure is should be elaborated on and its importance described, if it is to be included.

RESPONSE: This has been re-worded and clarified in the revised manuscript text and a specific reference has been added. The reworded sentence now reads:

'Utilizing data derived from cryo-EM technologies, the Scheres laboratory at the MRC Laboratory of Molecular Biology, Cambridge, UK have recently described a basic hierarchical classification of tauopathies that can be made on the basis and nature of the unusual folded structuring of tau filaments in each disease type [34].'

COMMENT: A little more discussion of why irresolvable and inert proteinaceous inclusions in the cell seem to be so harmful would be useful.

RESPONSE: These proteinaceous inclusions have been demonstrated by many laboratories to be both neurotoxic and extremely pro-inflammatory, especially over time. This aspect has been re-worded and clarified in the revised manuscript text and 3 specific references are now provided.

Please note that the ‘pro- inflammatory’ nature of both amyloid- and tau-bases inclusions have been mentioned several times in this ‘Commentary’ article.

COMMENT: A few statements consist largely of ‘filler” without actually moving things forward. For example “Our molecular and mechanistic understanding of the contribution of specifically misfolded cellular amyloid and MAPT proteins to AD and/or PrD neuropathology continues to evolve…”  Cutting down on these should be considered.

RESPONSE: We appreciate the Reviewers comments, however we imply exactly what we have stated. This is a ‘Commentary’ article written and presented for a wide-ranging scientific and medical audience interested in lethal neurological disease mechanisms and not just for prion and/or tauopathy experts. We maintain that this is still a largely unknown and unexplored area of neurology and neurobiology and our understanding of the neuropathology and disease mechanisms continues to evolve.

=========== 

END OF COMMENTS/RESPONSES